# A Narrative Review on Biochemical Markers and Emerging Treatments in Prodromal Synucleinopathies

**DOI:** 10.3390/clinpract15030065

**Published:** 2025-03-17

**Authors:** Jamir Pitton Rissardo, Ana Leticia Fornari Caprara

**Affiliations:** Neurology Department, Cooper University Hospital, Camden, NJ 08103, USA; fornari-caprara-ana@cooperhealth.edu

**Keywords:** alpha-synuclein, synucleinopathies, biofluid, prodromal, RBD, Parkinson’s disease, prevention

## Abstract

Alpha-synuclein has been associated with neurodegeneration, especially in Parkinson’s disease (PD). This study aimed to review clinical, biochemical, and neuroimaging markers and management of prodromal synucleinopathies. The prodromal state of synucleinopathies can be better understood with PD pathophysiology, and it can be separated into premotor and pre-diagnostic phases. The incidence of PD in patients with prodromal phase symptoms ranges from 0.07 to 14.30, and the most frequently studied pathology is the REM behavioral disorder (RBD). Neuroimaging markers are related to dopamine denervation, brain perfusion changes, gross anatomy changes, and peripheral abnormalities. α-synuclein assays (SAA) in CSF revealed high sensitivity (up to 97%) and high specificity (up to 92%); in the last decade, there was the development of other matrices (blood, skin, and olfactory mucosa) for obtaining quantitative and qualitative α-synuclein. Other biomarkers are neurofilament light chain, DOPA decarboxylase, and multiplexed mass spectrometry assay. Regarding genetic counseling in α-synucleinopathies, it is an important topic in clinical practice to discuss with patients with high-risk individuals and should involve basic principles of autonomy, beneficence, and non-maleficence. Some of the themes that should be reviewed are the involvement of physical activity, diet (including alcohol, coffee, and vitamin supplementation), smoking, sleep, and stress in the pathophysiology of synucleinopathies. The number of trials related to prodromal symptoms is still scarce, and the number of studies evaluating intervention is even lower.

## 1. Introduction

Parkinson’s disease (PD) is a progressive neurodegenerative disorder primarily characterized by motor symptoms such as bradykinesia, resting tremor, rigidity, and postural instability. These clinical manifestations are attributed to the degeneration of dopaminergic neurons in the substantia nigra pars compacta, leading to striatal dopamine deficiency. Beyond motor symptoms, PD encompasses a range of non-motor features—including hyposmia, constipation, rapid eye movement (REM) sleep behavior disorder (RBD), depression, and cognitive impairment—that often precede motor signs by years, indicating a prodromal phase of the disease.

PD is classified within the spectrum of synucleinopathies, a group of neurodegenerative disorders marked by the pathological accumulation of misfolded alpha-synuclein protein in neurons, glial cells, or nerve fibers. This group also includes dementia with Lewy body dementia (LBD) and multiple system atrophy (MSA), each presenting distinct clinical profiles yet sharing overlapping pathological features. In PD and LBD, alpha-synuclein aggregates predominantly in neurons, forming Lewy bodies and Lewy neurites, whereas, in MSA, the protein accumulates mainly in oligodendrocytes as glial cytoplasmic inclusions [1].

Historically, research has predominantly focused on the symptomatic treatment of PD during its clinical phase. However, over the past decade, there has been a paradigm shift towards understanding the prodromal stages of PD and other synucleinopathies. This shift aims to identify early biomarkers and therapeutic targets to facilitate disease-modifying interventions before the onset of overt clinical symptoms. Notably, markers such as olfactory loss, RBD, and constipation have been identified as significant predictors of prodromal PD [2].

In this review, we critically examine the current landscape of clinical, biochemical, and neuroimaging markers pertinent to the prodromal phases of synucleinopathies. We also discuss strategies for counseling and managing individuals identified as being in these early stages, with the objective of providing a comprehensive resource for movement disorder specialists and researchers dedicated to advancing the early diagnosis and intervention in synucleinopathies.

## 2. Clinical Markers of Prodromal Synucleinopathy

### 2.1. The Concept of Prodromal State in Parkinson’s Disease

The prodromal synucleinopathy in Parkinson’s disease (PD) is considered by many authors the first three stages from neurodegeneration to the clinical diagnosis, which involves the preclinical, premotor, and prediagnostic stages (Figure 1). In this context, it is important to define the prodromal state for the discovery of neuroprotective therapy that may lead to the cure of this neurodegenerative condition.

The risk phase involves genetic and environmental susceptibility without detectable pathology. The pre-clinical phase features neurodegeneration without overt symptoms. Also, the pre-clinical phase of Parkinson’s disease is more theoretical as manifestations of this phase tend to not be clinically detectable. There are no cohort studies, and the natural course and timeline are unknown [3]. Moreover, the initial of the preclinical phase is characterized by the inflection point between the noradrenergic [4] and dopaminergic loss [5].

The prodromal phase presents subtle non-motor symptoms, such as hyposmia and REM sleep behavior disorder, preceding motor dysfunction. There are several well-designed cohort studies with prodromal clinical markers for Parkinson’s disease (Table 1). Some examples are population-based studies with rapid eye movement sleep behavior rapid eye movement (RBD) cohort studies, such as PREDICT-PD, Parkinson Associated Risk Study (PARS), Prospective Validation of Risk factors for the development of Parkinson Syndromes (PRIPS), Tuebinger evaluation of Risk factors for Early detection of NeuroDegeneration (TREND), and the North American prodromal synucleinopathy (NAPS). Some of the symptoms evaluated by these cohorts were olfaction loss, constipation, autonomic symptoms (orthostatic dysfunction, dizziness, urinary incontinence), anxiety, depression, sleep impairment (excessive daytime sleepiness, dream enactment behavior), visual symptoms (dimness, blurring, night-blindness, color impairment, decreased contrast), subtle motor signs and cognitive symptoms (forgetfulness, memory complaints). The tests performed in these cohorts were neuropsychological tests (MMSE, MoCA, and full cognitive battery), quantitative motor tests (UPDRS), and REM sleep without atonia severity [6].

Mahlknecht et al. performed a data-driven approach to a definition of prodromal PD using Bayesian naive classifier methodology, in which the author evaluated the Movement Disorder Society (MDS) criteria for prodromal PD. The sample size for a randomized control trial with a year treatment duration, an effect size of approximately 20%, and a power of 90% required more than 1226 individuals for each group [12]. Another possible limitation in prodromal studies is that the evolution of markers is not sufficiently clear in prodromal stages, only motor symptoms can be detected to be significantly increasing after the diagnosis. Additionally, cognitive markers tend to be difficult to assess. For other non-motor symptoms, there are significant overlaps and confounding symptoms in the elderly population [18]. Therefore, other symptoms need to be used to assess phenoconversion reliably in clinical trials.

### 2.2. Isolated Rapid Eye Movement Sleep Behavior Disorder and Synucleinopathy

RBD and PD share a common molecular pathology centered on α-synuclein misfolding and aggregation. In RBD, early deposition of phosphorylated α-synuclein occurs in brainstem nuclei, particularly the sublaterodorsal nucleus and magnocellular reticular formation, leading to dysfunction in REM sleep atonia pathways. This pathology spreads via prion-like mechanisms, progressively affecting dopaminergic neurons in the substantia nigra pars compacta [19]. At the cellular level, α-synuclein aggregates impair mitochondrial function, disrupt endoplasmic reticulum homeostasis, and activate neuroinflammatory pathways through microglial and astrocytic activation. Dysregulated protein degradation via the ubiquitin–proteasome system and autophagy–lysosomal pathways exacerbates neuronal stress, contributing to progressive neurodegeneration [20]. Additionally, tau co-pathology and lysosomal dysfunction further accelerate disease progression. Identifying molecular signatures of α-synuclein pathology in RBD may facilitate early diagnosis and neuroprotective interventions [21].

Overall, the annual rate of phenoconversion of RBD to parkinsonian syndrome in the Seoul National University Boramae Medical Center (SNU-BMC) RBD cohort was 7% [22] and in the international RBD study group was 6–8% [23]. The specificity of RBD for synucleinopathy was reported to be more than 90% independent of the occurrence of other features [24]. In this way, RBD may be a very suitable population for phenoconversion in which trials evaluating neuroprotective substances should be attempted.

It is estimated that abnormal olfactory function, impaired color vision, and mild motor dysfunction increase the likelihood of neurodegeneration in RBD by up to 200% within five years of the disease. The motor symptoms were measured by the Unified Parkinson’s Disease Rating Scale (UPDRS), had an odds ratio of around 3, and the olfactory abnormality with 2.6. The neuropsychological testing in patients with mild cognitive impairment (MCI) revealed an odds ratio (OR) of 2.37 [25]. In a study comparing the phenoconversion of RBD to PD and Lewy body dementia (LBD), it was noticed that the most important factors for phenoconversion to PD are motor signs, olfaction, and constipation; instead, for phenoconversion to LBD are color, cognition, and olfaction, cognition [26]. Also, motor scores tend to progress faster and require lower sample sizes, but cognitive, olfactory, and autonomic variables showed modest progression with high variability [27]. In this way, a three-year trial study evaluating motor symptoms requires only approximately 100 participants to show the efficacy of therapies in phenoconversion. Therefore, it is possible that the motor exam is one of the most reliable clinical markers in monitoring disease progression in prodromal Lewy body disease (pro-LBD).

Patients with RBD also begin to present progressive cognitive impairment a decade before phenoconversion, in which attention and executive function are likely the strongest predictors for LBD and episodic memory to PD [28]. In this way, a decline in cognitive function in RBD can be detected very early in the disease course. Also, the pattern of cognitive dysfunction begins with attention and executive dysfunctions, followed later by dysfunction in the memory. Moreover, the presence of constipation can be a predictor of faster cognitive and motor decline in RBD [22]. This finding highlights the possible role of the gut–brain axis in disease progression in prodromal synucleinopathies.

#### The Evolving Concept of the Prodrome in REM Sleep Behavior Disorder

Stefani et al. reported electromyographic activities during REM sleep in a subset of patients with and without atonia, and they found that REM sleep without atonia (RWA) is associated with phenoconversion to RBD [29]. Another study reported that individuals with dream-enactment behaviors (DEB) but with subthreshold RWA, eventually developed RBD (phenoconversion rate was 66%) and had a significant increase for neurodegenerative disorders (2.95 risk increase) after eight years of follow-up [30]. The authors also suggested that two types of prodrome RBD may exist. One type is characterized by full-blown RBD, while another is milder, exhibiting less pronounced features, or represents late converters, where neurodegeneration becomes apparent only at long-term follow-up.

In a Korean population-based cohort study, the authors defined prodromal RBD as isolated RWA and any electromyography (EMG) activity of more than twenty percent without repeated DEB. The authors found that the prevalence of RBD was 1.4%, isolated RWA 12.5%, and isolated DEB 3.4%. However, the data to assess phenoconversion or the associated factors to evolution to RBD were poor (positive predictive value of 7.7%), and further studies are definitely needed [31].

RWA was already associated with the development of RBD [32], and the clinical use of dopamine transporter imaging [33] and brain MRI with voxel-based analysis [34] can support further this finding. In this context, the severity of RWA can predict the development of PD, in which increased tonic chin with EMG recording during REM sleep was revealed to be correlated with phenoconversion [35].

### 2.3. Prodromal Markers of Multiple System Atrophy

There are few studies on prodromal multiple system atrophy (MSA). In patients with primary autonomic failure (PAF), there are features related to phenoconversion to MSA and LBD. Noteworthily, 23–43% of the patients with MSA develop autonomic disorders as the first manifestation, which are characterized by mild degree of cardiovagal impairment, preganglionic pattern of sweat loss, severe bladder dysfunction, and supine norepinephrine > 100 pg/mL. In cases of phenoconversion to PD/LBD, abnormalities in motor signs and a pronounced lower norepinephrine plateau (greater than a 65 pg/mL difference from healthy controls) were statistically significant [36]. Another study reported that olfactory score, supine norepinephrine, and heart rate response in patients with PAF are the three features to differentiate isolated PAF from phenoconversion to MSA and PD/LBD [37].

In contrast to the PAF population, the prodromal MSA evolving from RBD is quite different. A case series of MSA evolved from RBD revealed that motor progression is fast, uncommon orthostatic hypotension, bladder symptoms may precede orthostatic symptoms, neurocognitive tests are usually normal, and sleep patterns are the same. However, the olfactory function remained an important marker to differentiate the phenoconversion [38].

### 2.4. Prodromal Criteria by the Movement Disorder Society

The frequency of MSA phenoconversion using the prodromal MSA by the MDS is 6–8% among RBD and 8–28% among individuals with PAF. Also, the predictors for phenoconversion were supine norepinephrine > 100 pg/mL, preserved olfaction, supine heart rate > 70 bpm, age of onset in the early 50s, orthostatic heart rate increase of 10 bpm within 3 min, composite autonomic severity vagal score < 2, preganglionic sweat loss pattern, and subtle motor signs not qualifying for parkinsonism or ataxia at initial assessment [39]. However, there are no consensus criteria for prodromal MSA with other types of presentations besides those with RBD and PAF.

In 2015, the Movement Disorder Society proposed the research criteria for prodromal PD [40] and, in 2022, for prodromal MSA [39]. The criterion for prodromal PD is a calculation of an estimated probability of more than 80% of the patients developing PD. On the other hand, the prodromal MSA is based on specific criteria of inclusion and exclusion, which can lead to significant variability in the apparent percentages of phenoconversion (Figure 2).

## 3. Neuroimaging Markers in Prodromal Synucleinopathy

For the neuroimaging markers in prodromal PD/LBD, there are many imaging studies with well-designed cohorts with longitudinal data that are reliable tests to detect individuals with prodromal states (Table 2). However, few neuroimaging studies for prodromal MSA were completed.

### 3.1. Imaging Nigral Dopaminergic Change in Prodromal Parkinsonian Syndromes

Dopamine loss in PD is not uniform; it involves distinct nigrostriatal, mesocorticolimbic, and tuberoinfundibular pathways, each contributing to specific motor and non-motor symptoms. In the early stages, PD primarily affects dorsal striatal dopaminergic terminals, particularly the putamen, leading to bradykinesia and rigidity due to impaired D1/D2 receptor-mediated modulation of the direct and indirect basal ganglia pathways [41]. As the disease progresses, ventral striatal dopamine loss disrupts reward processing, contributing to apathy and depression. Degeneration of mesocortical dopamine neurons projecting to the prefrontal cortex impairs executive function and working memory, while alterations in the tuberoinfundibular pathway contribute to hyperprolactinemia and autonomic dysfunction [42]. Additionally, differential loss of tonic versus phasic dopamine release exacerbates motor fluctuations and cognitive deficits [43].

Dopaminergic and substantia nigra imaging studies in RBD reported abnormalities frequently found in PD, but with lesser degree or lower frequency compared to PD. In this context, RBD was already associated with decreased striatal dopamine in DAT scans [44], increased posterior substantia nigra free water [45], loss of nigrosome 1 signal [46], and decreased nigral neuromelanin [47].

From early PD studies like the Parkinson Progression Marker Initiative (PPMI), the annual decline of dopaminergic markers was estimated to be up to five percent. A relatively more rapid decline is expected to occur during the prodromal phase when the early PD data are extrapolated [48]. Pathological studies suggested a threshold for parkinsonian motor symptoms manifestation after 40–60% cell loss in the substantia nigra pars compacta and 60–70% in the striatal dopamine depletion. This data suggests that a reliable marker of prodromal PD/LBD could be the level of dopaminergic reduction ranging from 45 to 75% of normal values [49].

In N-3-[18F]fluoropropyl-2β-carbomethoxy-3β-4-iodophenyl nortropane positron emission tomography (18F-FP-CIT PET) imaging, there are some soft striatal changes in RBD patients when compared to healthy controls. Also, there are two types of RBD groups, one with normal uptake and the other with decreased (25–50% dopamine loss) uptake; also, the hypodopaminergic compared to the normodopaminergic group had a high risk of phenoconversion [50]. In addition, longitudinal studies in Barcelona [51] and Korea [52] revealed that RBD with more than 25% reduction of dopamine in the posterior putamen has a high phenoconversion rate during the follow-up. Furthermore, rapid dopamine transporter imaging loss in RBD was correlated with abnormal olfactory function and multiple prodromal markers [52]. In the Parkinson-associated risk study, only individuals with significant levels of dopamine loss in the prodromal range showed a high rate of phenoconversion and a faster dopamine loss, and the annual decline was 6.7% per year in hyposmic subjects with initial dopamine reduction of 35% [53].

A longitudinal multimodal neuroimaging study investigating the dopamine transporter, nigral neuromelanin, and quantitative susceptibility mapping in RBD and PD aimed to track neurodegenerative changes over time and better understand the progression from RBD to PD. The study revealed that dopaminergic loss occurs first in the sensory–motor area, followed by the limbic system, and associative areas; on the other hand, the first abnormalities in neuroimaging concerned dopamine transporter, followed by quantitative susceptibility mapping and neuromelanin loss [54].

The locus coeruleus is the responsible region for RBD, and imaging studies have shown neuromelanin loss in the locus coeruleus [55]. Previous evidence suggests that the locus coeruleus neuromelanin loss precedes nigral dopamine loss in RBD [56]. However, the relative risk of phenoconversion or disease progression in individuals who have isolated locus coeruleus changes (without striatal dopamine abnormalities) is not known [57].

### 3.2. Peripheral Autonomic Denervation

A multimodal colonic, cardiac, and dopamine imaging study revealed that the first abnormality to occur in patients with phenoconversion is impairment in cardiac sympathetic activity, followed by abnormal colonic function, and lastly the loss of dopamine in the striatum [58]. Nevertheless, these results should be cautiously evaluated because the sample of the study involved only 12 individuals.

Both CNS and PNS are involved in incidental LBD. In this context, studies found phosphorylated α-synuclein in different tissues including the submandibular glands [59] and in the retina [60]. Interestingly, the brain synucleinopathy score is correlated with synucleinopathy scores of both regions: the submandibular gland and retinal. Also, an experimental study revealed that α-synuclein aggregations can propagate from the retina to the brain [61]. Moreover, a study assessing the retina and optic nerve in various neurodegenerative conditions showed that α-synuclein pathology is present significantly in the retina and optic nerve of only individuals with LBD or MSA, but not in PD and PD dementia [62].

Optical coherence tomography (OCT) detects retinal thinning in early PD and RBD. An interesting finding in these syndromes was the retinal involvement including the ganglion cell complex thinning that correlated to olfactory loss and striatal dopamine transporter reduction [63]. An OCT angiography study reported retinal thinning and deep capillary hypervascularization in patients with RBD [64]. Also, there are studies showing retinal or retinal nerve fiber layer thinning in MSA, but no imaging study reported on “prodromal MSA”.

A large cohort from the UK Biobank was studied to identify early features of Parkinson’s disease (PD) using optical coherence tomography (OCT). The findings revealed an association between inner retinal layer thinning and the progression to PD [65]. So, this suggests the potential of OCT imaging as a tool for triaging high-risk individuals with prodromal symptoms.

### 3.3. Metabolic Network Activity

The use of 18F-fluorodeoxyglucose (FDG) positron emission tomography (PET) in RBD, combined with singular spectrum analysis and principal component analysis, allows for the identification of distinct patterns of metabolic activity. This approach can offer a digital representation of the core expression patterns in each individual, enhancing the understanding of RBD and its progression. In PD, there was higher uptake of FDG in the caudate, putamen, and thalamus; the other areas with uptake were the cerebellum and decreased uptake in the premotor and posterior parietal cortex [66]. In cases of RBD, the uptake was stronger in the hippocampus, putamen, globus pallidus, thalamus, and cerebellum with decreased uptake in occipital and occipitotemporal areas [67]. The expression of both patterns (mixture of patterns) is known as a “de novo” PD-RBD-related pattern (RP), and it has a high rate of phenoconversion to LBD [68]. Additionally, “de novo” PD-RBD and LBD-RBD RPs have been consistently observed in independent RBD samples from Italy and Korea. These patterns showed no significant difference in predicting phenoconversion from PD or LBD-RBD RPs. This suggests that the two phenotypes of LBD associated with RBD may represent a spectrum of a common condition, at least in terms of the brain’s metabolic pattern [69]. Furthermore, another study from Italy discovered an RP to phenoconversion in patients with RBD [70], and the same pattern was able to be reproduced in a study from Korea [71]. 

The reproducibility of metabolic patterns in predicting phenoconversion in different RBD populations is statistically and clinically significant for PD RP, “de novo” PD-RBD RP, “de novo” LBD-RBD RP, and RBD conversion RP. Interestingly, to the authors’ knowledge, the FDG PET-related patterns of RBD are the only ones observed to be reproducible in different populations. The predictability of FDG RP to phenoconversion is up to eight, which is better than visual analysis of dopamine transporter imaging, and slightly higher than semiquantitative analysis of the same imaging technique. However, there is poor discriminative value when comparing PD versus LBD conversion with an AUC of approximately 0.6 [71].

MCI is prevalent in the RBD population, which is even higher than in PD. Many different imaging studies consistently reported abnormalities in the posterior brain regions associated with cognitive impairment in RBD [72]. Some authors even hypothesized a hub of a functional network subserves in the cuneal/precuneal area as a neuroimaging marker for cognitive impairment in individuals with RBD [73]. Also, occipital glucose hypometabolism may predict phenoconversion, which was a common feature observed among RBD, PD dementia, and LBD [74]. Noteworthily, unclear cerebellar hypermetabolism that increased over time was already observed in association with occipital hypometabolism [75].

### 3.4. Brain Atrophy

Interestingly, RBD patients showed typical patterns of cortical atrophy which was consistently reported in many different brain MRI studies [76]. Some patients can have cortical and subcortical involvement of the gray matter. This characteristic was more pronounced in individuals reporting memory complaints [77]. Also, the cortical layer thickness in RBD has been associated with phenoconversion to LBD, highlighting its potential as an early biomarker for predicting disease progression [78]. Besides hypometabolism in the occipital region, the decreased length of the cortical layer in this area and the orbitofrontal region was related to RBD progression [79]. Another study found an LBD pattern in some RBD patients with increases over time and they later phenoconverted to LBD, also a progressive reduction in the cortical thickening over four years follow-up in RBD patients, which predicts a phenoconversion [80].

A multicenter study found a prodromal LBD signature using deformation MRI data and clinical variables in RBD patients, and this pattern was elevated in LDB, prodromal LDB, and LDB-RBD phenoconverters [81]. Also, this study suggests that imaging markers can detect changes earlier than clinical manifestations, offering the potential for earlier diagnosis and intervention in neurodegenerative diseases like RBD and LBD.

The investigation of the cortical and subcortical gray matter atrophy in RBD patients led to the discovery of amygdala, gyrus rectus, and olfactory cortex atrophy, and, these patients were more likely to phenoconvert to LBD [82]. These observations are in line with a study investigating central cholinergic progressions through imaging, in which patients with LBD compared to PD had severe occipital denervation with more profound denervation in the limbic than the occipital cortex [83]. This study suggests that relative cholinergic loss can discriminate between PD and LBD. Moreover, it is possible that future studies may show that central cholinergic loss in RBD is faster in PD, but it can be more pronounced in LBD. Also, the central cholinergic atrophy was associated with the development of LBD. When analyzing both the cholinergic and dopaminergic systems simultaneously, some individuals with LBD-phenoconversion and others with PD-phenoconversion may exhibit shared features, suggesting overlapping neurobiological pathways between these conditions [22,82].

### 3.5. Amyloid Imaging

Amyloid pathology also plays a role in LBD. A study revealed an increase in the percentage of abnormal β-amyloid PET in RBD (25%), MCI-RBD (41%), and LBD (60%) compared to control (19%). But, there was no clear relationship between the clinical signs (or even severity) and beta-amyloid imaging [84]. By amyloid tracer PET imaging analysis, the cortical uptake was evaluated in RBD patients including phenoconverters, and it was found that TMT-B performance (earliest and cognitive marker) was linearly correlated with amyloid uptake [85].

## 4. Biofluid Markers in Prodromal α-Synucleinopathies

During the three stages of preclinical, premotor (prodromal), and pre-diagnostic stage, there is a window of opportunity to measure biomarkers with objective clinical measurements, neuro-imaging, and laboratory-based analysis of tissues and biofluids [86]. Also, the concept of staging classification, which defines the progression of the disease, is valuable for enrolling patients earlier in clinical trials [87]. For example, the diagnosis today of PD depends on motor symptoms, but the disease process in itself can start more than 20 years prior with α-synuclein pathology (CSF) and dopaminergic degeneration (dopamine transporter imaging). Also, an example of early disease trials is the path-to-prevention trials (P2P) in PD.

### 4.1. α-Synuclein Assays and Matrices

The concept of seeding aggregation assays (SAA: PMCA/RT-QuIC), involves introducing recombinant monomeric α-synuclein into a CSF sample. As aggregates present in the CSF sample seed the aggregation of the monomer, this process can be detected by the SAA. The aggregation leads to a detectable peak in the signal, providing a sensitive method for identifying α-synuclein aggregation in the CSF (Table 3). SAA in CSF revealed high sensitivity (up to 97%), high specificity (up to 92%), and high congruence across different laboratories [88]. Interestingly, these results are consistent independently of the technique PMCA or RT-QuIC [89].

Some patients with PD are SAA negative, and similar results have been observed in individuals with LRRK2 mutations and moderate-to-high scores on the University of Pennsylvania Smell Identification Test (UPSIT). This suggests that α-synuclein aggregation, as detected by SAA, may not be present in all PD patients, including those with specific genetic mutations or sensory changes [99]. So, further studies are needed to assess the reason for these patients with PD of up to 15% having negative SAA results. Interestingly, there are different peak points to MSA (100 h) and PD (200–250 h) that can suggest different seeding dynamics between these two disorders in PD versus atypical PD and different conformational strains of α-synuclein [101]. A qualitative comparison between 150 h and 24 h CSF α-synuclein SAA in PD, LBD, RBD, MSA, and non-synucleinopathies revealed a different peak up for PD that is high at 12 h and for MSA low in 9 h [102]. It is worth mentioning that these results were confirmed by neuropathological analysis.

α-synuclein is abundantly present outside the central nervous system and was found in the postmortem and living patients in different locations. In postmortem studies, α-synuclein was located in the stellate ganglion, paravertebral sympathetic ganglia, vagus nerve, epicardial plexus, mesenteric sympathetic ganglia, enteric nervous system, adrenal gland, and genitourinary tract. On the other hand, in living individuals, it was found in minor salivary glands, submandibular glands, stomach, colon, and skin [103]. The skin is probably the best peripheral matrix for SAA, and it has already been studied in RBD subjects [97].

There are two main techniques for preparing blood matrix samples for SAA. One involves immunoprecipitation, where specific antibodies are used to isolate target proteins, such as α-synuclein, from the blood sample before performing the assay. The other technique focuses on extracting extracellular vesicles, which can carry aggregated α-synuclein, from the blood sample prior to conducting the SAA. Both methods aim to enrich the relevant components for more sensitive detection of protein aggregation. However, studies implementing these techniques are scarce (Table 4). Interestingly, the extracellular vesicles as a serum marker of α-synuclein were already obtained from individuals with high-risk PD, and the total α-synuclein level was found to be increased in RBD and individuals with high-risk PD [104].

Another possible marker of α-synucleinopathies is the L1 cell adhesion molecule (L1CAM)-enriched extracellular vesicles (L1EVs). This biomarker is obtained through the measurement of α-synuclein in the extracellular vesicles which are membrane-containing structures with the ability to cross the blood–brain barrier (Table 5). However, there are some limitations related to this method such as the unspecific source of α-synuclein with peripheral (serum) measurement, unspecified amount of α-synuclein encountered in the L1EVs, and nonspecific signals of L1EV comparing the source from central and from peripheral system.

### 4.2. Other Biomarkers

#### 4.2.1. Neurofilament Light Chain Concentration

The neurofilament light chain (NfL) is a structural protein found in neurons, specifically in the axonal cytoskeleton. Neurofilaments are released from degenerating neurons, then accumulate in the cerebrospinal fluid and eventually end up in peripheral blood. Although NfL is not specific to Parkinson’s disease (PD), it exhibits a characteristic slow and progressive increase over time. Notably, the peak NfL level is lower in the early stages of PD compared to other neurodegenerative disorders, where the peak is typically higher. This distinct pattern of NfL accumulation may provide insight into disease progression and help differentiate PD from other conditions. The concentration of NfL can be used for prodromal evaluation, as indicated by the progression curve. However, the study fails to provide statistically significant data to support its use as a reliable biomarker for early detection or disease progression [109].

#### 4.2.2. DOPA Decarboxylase

DOPA Decarboxylase (DCC) catalyzes catecholamines and it has already been found in different matrices for different pathologies including PD, LBD, and atypical PD. Initially, it was hypothesized that high levels of DCC were related to medications used in the treatment of PD. However, later studies found that elevated DCC levels were also present in patients with neurodegenerative conditions, even in the absence of any treatment. This suggests that increased DCC levels may be associated with the neurodegenerative process itself, rather than being solely influenced by PD medications [110]. In individuals with prodromal PD, DCC was found to be normal in the blood and high in the CSF [111].

#### 4.2.3. Multiplexed Mass Spectrometry Assay

A proteomic assay first untargeted and then targeted with independent validation identified proteins that are suitable for the current pathways related to prodromal PD, with upregulation (C3, SERPINA 3) and downregulation (GRN, DKK3) of proteins related to inflammation and neuroprotection, respectively. Moreover, the authors found a panel able to predict phenoconversion in patients with RBD in 79% of the cases with eight blood proteins (NCBI Gene identification numbers were 2896, 10,747, 3309, 5730, 3383, 718, 27,122, 710) seven years before PD diagnosis [112].

### 4.3. Gut Microbiome

The gut microbiome plays a crucial role in PD pathogenesis via the gut–brain axis, influencing α-synuclein aggregation, neuroinflammation, and intestinal permeability. Dysbiosis disrupts short-chain fatty acid production and increases lipopolysaccharides, exacerbating neurodegeneration [113]. PD-associated gut microbiota alterations include reduced Prevotella and Bifidobacterium, alongside increased Enterobacteriaceae, Desulfovibrio, and Akkermansia, promoting gut inflammation, α-synuclein misfolding, and dopaminergic neuronal loss [114]. These findings highlight the gut microbiome’s role in PD progression and its potential as a therapeutic target.

Akkermansia species is commonly found in the gut microbiome of patients with PD. It is believed that this bacteria digested the mucous barrier leading to increased absorption of toxic substances and causing inflammation in the periphery culminating in triggers of autoimmune disorder, cancer, metabolic, and neurodegenerative conditions [115]. Also, the dysbiosis in the gut can lead to the secretion of metabolites of these bacteria promoting a “leaky” gut, which allows the systemic spreading of amyloidogenic peptides (curli protein produced by *E. coli*) [116]. Interestingly, a high-fiber diet can protect the neurological system by avoiding bacterial integration and avoiding conversion to curly synuclein in the enteric system [117]. Moreover, certain pathophysiological processes, such as the amyloidogenic process observed in the gut, may act as triggers for the development of PD. Similar mechanisms involving protein aggregation in the gut could contribute to the systemic spread of amyloidogenic peptides, potentially initiating α-synuclein aggregation in the brain and triggering the onset of PD (https://www.michaeljfox.org, accessed on 5 February 2025). α-Synuclein can also be found in PD with nasal washing and quantification of the total level of this protein.

### 4.4. Parkinson Progression Marker Initiative and Path-to-Prevention Studies

The Parkinson Progression Marker Initiative (PPMI) regarding fluid biomarkers started their study in 2024. One of the objectives of the PPMI study is to identify additional biomarkers, beyond SAA in CSF, to improve the screening of neuronal synucleinopathies. Also, there is also growing interest in the co-pathology observed across various studies, as well as the identification of different biological subphenotypes, to better predict phenoconversion. Understanding how these co-pathologies and subphenotypes interact may provide valuable insights into the progression of neurodegenerative diseases and help identify individuals at risk of transitioning to more advanced stages of conditions like PD or LBD (Figure 3) (https://www.ppmi-info.org, 5 February 2025). The PPMI study will be looking at variable available platforms using Olink® (plasma and CSF) and NuLISA® (plasma and CSF). The PPMI core biomarker is α-synuclein SAA in the CSF, but other targeted neurodegenerative markers will be analyzed such as NfL (plasma), GFAP (plasma), p-tau217 (plasma), AB40/42 (CSF), and p-tau181 (CSF).

The Path-To-Prevention (P2P) study enrolls individuals in the prodromal stages of PD to do platform trials to compare different compounds and highlight individuals at risk of developing PD (https://www.ppmi-info.org/study-design/path-to-prevention-platform-trial, accessed on 5 February 2025). The launch is programmed for the fourth trimester of 2025, but the start-up and development began in 2024.

## 5. Counseling of Prodromal Symptoms in α-Synucleinopathies

### 5.1. Basic Principles of Early Risk Disclosure

Patients may present with prodromal symptoms as their chief complaint, or they may be identified incidentally during routine screenings or other medical evaluations. There are two fundamental limitations of early risk disclosures. The first is that there is no clear advantage from the perspective of treatments since there is no evidence-based intervention that might prevent or slow down the progression of the disease. Second, there is uncertainty regarding if and when PD will be diagnosed, as the data are often based on non-specific prodromal symptoms, which can be shared with other conditions. Furthermore, genetic risk variants or mutations associated with PD may exhibit incomplete penetrance, meaning that not all individuals with these genetic factors will develop the disease. This adds complexity to early diagnosis and risk prediction, as some individuals may never show clinical signs despite having genetic predispositions [118]. Also, it is worth remembering that basic principles for risk disclosure involve respect for autonomy, beneficence, and non-maleficence. In regards to autonomy, informed decision-making indicating the potential advantages and disadvantages of receiving a diagnosis should be disclosed. Moreover, if the patients are being enrolled in clinical trials, the disclosure of results and transparency regarding the reasons for patient enrollment is strongly recommended, ensuring that participants fully understand the purpose and potential outcomes of the study. In cases where ethical considerations are involved, the inclusion of an ethicist in the decision-making process is advisable to uphold the integrity and ethical standards of the research.

#### 5.1.1. Respect for Autonomy

Only 55% of patients with RBD and 12% of those with RWA received prognostic counseling regarding the potential risk of phenoconversion. The physicians that most commonly disclosed information were male sleep neurologists, and another was elderly [119]. Another study from the UK with RBD patients revealed that only a small percentage (1/3) was not informed regarding phenoconversion, and many patients (72.2%) searched for this information online [120]. Also, most of the sleep medicine experts (93.2%) counseled patients with RBD on phenoconversion prognosis, and some (15.9%) provided routine details regarding phenoconversion estimated risks, but only 31.8% asked the patient preference in knowing the risks of phenoconversion [121]. Interestingly, patients with RBD had a strong preference for receiving prognostic information about neurodegenerative diseases because they found it personally important (92.5%), important to maintain trust in their physician (87.6%), or even because they desire to know more information (95.7%) to discuss future life planning and future neuroprotective therapies; however, 9.9% of the patients with RBD prefer to be inquired before the disclosure of prognostic information [122].

A study assessing patients with PD and RBD reported that most (64%) physicians believe that their patient wants to know of the high risk of developing PD once diagnosed with RBD. And, this belief was supported by most patients (69.2%) reporting that they want to know about the risks of phenoconversion [123]. Noteworthy, patients with prodromal symptoms compared to those with neurodegenerative conditions tend to more commonly ask for counseling and information because they want to attempt neuroprotective measures to decrease their risk of developing the condition. Also, the study found that many individuals do not want to receive the diagnosis because of psychological impact including lower mood and quality of life, powerlessness, and uncertainty of the diagnosis. Moreover, the authors found that many patients have high expectations of the health system with regular follow-up visits (100%), enrollment in clinical trials (92.3%), recommendations regarding lifestyle (87.2%), information about PD (64%), and additional support (69%).

Some patients (39%) with PD also reported that they would change something in their lives if they knew about the risk of phenoconversion [124]. Another study from Turkey revealed a similar finding (32%); also, it revealed that females more frequently act following risk disclosure and patients with family history of PD are more likely to favor disclosure [125].

#### 5.1.2. Beneficence and Non-Maleficence

Risk disclosures have both advantages and disadvantages. The advantages are future care and life planning, enhanced patient–physician relationships, increased participation in neuroprotective trials, and early symptomatic treatment. On the other hand, the disadvantages are anxiety and distress, long latency periods, probabilistic not deterministic, and lack of disease-modifying agents [126]. To maximize the benefits, the physician should offer regular follow-ups, provide prognostic counseling followed by the effect of lifestyle change, and provide education materials and contact information for the patient. To minimize the distress, the physician needs to assess psychological readiness and behavioral medicine referrals.

In a study assessing four weeks after genetic disclosure, most patients were not upset, sad, anxious/nervous, having problems enjoying life, and feeling regret. But, they were often frustrated with no available guidelines for definitive prevention [127]. The data from an Alzheimer’s study in disclosing amyloid PET results can be extrapolated to the PD community (Table 6), and the most important learning point was that there was no sustained difference in depression and anxiety.

#### 5.1.3. Risk Disclosure Flow

Before genetic disclosure, the physician should know the legislature surrounding this theme in his/her own country. Individual factors such as psychiatry history, support network, age, and educational background should be assessed. Also, the physician should ask open-ended questions and inquire if the patient wants to know the results and the prognosis, “would you be willing to know more about the association of sleep disorder, RBD, with other neurological conditions?” or “RBD may have implications for long-term health, do you want to know more about it [133]?” If the patient disagrees, consider providing a general statement like “would you like to be asked about that in the future?” The term neurodegenerative conditions should be avoided, instead Parkinson’s disease and dementia should be used.

## 6. Management of Prodromal Symptoms in α-Synucleinopathies, Evidence-Based Advice to Patients

### 6.1. Physical Activity

Physical activity is defined by MET (metabolic equivalent of task), a moderate-intensity exercise has 3–5 MET and a vigorous-intensity exercise has ≥6 MET. Physical activity can lower the velocity of PD progression, in which significant results were found with the highest level of total physical activity (relative risk 0.79), moderate to vigorous physical activity (relative risk 0.71), highest levels of total physical activity (male) (0.68), and moderate-vigorous physical activity (male) (0.68). Also, a dose–response was found with each increase of 10 MET hours/week decreasing the risk of PD by 10% in males [134]. Another study assessing specifically women with PD revealed a decrease of 25% in PD progression with the highest levels of total physical, and a dose–response was also noticed [135]. However, a specific level of exercise needed to achieve a benefit in decreasing the risk of PD is not known, the previous study showed a variation from 6.2 to 70.8 MET hours/week. The World Health Organization (WHO) provided some recommendations in the last guideline for adults living with disability in 2020 (Figure 4) [136].

### 6.2. Diet

Two diet patterns that were frequently studied in neurodegenerative conditions are the Mediterranean diet and the MIND diet (Mediterranean-Dietary Approaches to Stop Hypertension Intervention for Neurodegenerative Delay) (Table 7). The Mediterranean diet emphasizes vegetables, fruits, whole grains, legumes, olive oil, nuts, seeds, fish, and minimal red meat, with a focus on fresh, unprocessed foods, moderate portions, and flavorful herbs over salt. The MIND diet combines Mediterranean and Dietary Approaches to Stop Hypertension (DASH) diets, targeting brain health by focusing on 10 brain-healthy foods: leafy greens, other vegetables, berries, nuts, whole grains, beans, fish, poultry, olive oil, and wine. Some studies with Mediterranean and MIND diets showed a protective factor, but others did not show significant results. More than only having an effect on the general metabolism of glucose and lipids, it is possible that diet had a significant role in the microbiota leading to indirect effects due to the gut–brain axis. The literature is still scarce regarding ketogenic and other diet types [137]. Therefore, The MIND and Mediterranean diets may reduce neuroinflammation, and enhance gut microbiome balance, potentially slowing Parkinson’s disease progression and improving cognitive and motor outcomes.

### 6.3. Regarding Individual Food, Food Groups, or Nutritional Supplements

There are conflicting results in the literature regarding alcohol consumption and the risk of PD. A factor probably contributing to this is high heterogeneity and lack of adjustment for key confounding factors. The NIH-AARP Diet and Health study revealed that total alcohol consumption is not related to PD risk [153]. A meta-analysis with 11 prospective studies found that a higher intake of alcohol was associated with lower PD risk (RR 0.81), but these findings were only correlated with beer consumption (not liquor and wine) and Asian ethnicity (not North American or European) [154]. Another meta-analysis with 52 observational studies showed OR 0.84 in patients with alcohol intake and the risk of PD, but the studies were highly heterogeneous (I2 = 93.2%) [155]. Noteworthily, there is no indication to suggest patients drink alcohol due to alcohol addiction, increased risk of certain cancers, liver disease, cardiovascular disease, and intentional injuries (suicide, accidental injury, and death).

There is robust evidence suggesting that dairy consumption, particularly low-fat milk, may increase the risk of developing PD. In contrast, yogurt and ice cream have been associated with potential protective effects against PD. The impact of cheese on PD risk remains unclear, with no conclusive evidence either supporting or refuting its role in disease development. Milk is a nutritious product, rich in vitamins B and D, protein, and calcium, which are essential for overall health. However, the relationship between the overall consumption of dairy products and PD risk remains unclear (Table 8). Several hypotheses have been proposed to explain the association between dairy consumption and PD risk. These include environmental contaminants, such as pesticides and organochlorines, present in dairy products; excess calcium, which may decrease vitamin D levels and promote neurodegeneration; and reduced uric acid levels, which could contribute to oxidative stress. Therefore, dairy consumption may contribute to increased Parkinson’s disease risk through mechanisms involving oxidative stress, neurotoxic contaminants, and gut microbiota alterations. While epidemiological studies suggest an association, further research is needed to establish causality and underlying biological pathways.

Coffee and caffeine in general lowers PD risk. A meta-analysis with 16 case–control studies found a non-linear relationship between coffee and PD risk (RR 0.72), and a linear relationship between caffeine and tea with PD risk; also, the effect observed was stronger in males [167]. Another meta-analysis with nine healthy cohorts found that caffeine consumption had a significantly lower risk of PD (HR 0.79) [168]. Moreover, caffeine is known to reduce the risk of obesity, cardiovascular disease, type 2 diabetes mellitus, Alzheimer’s disease, cognitive impairment, and dementia; also, it can improve lipid profiles, fatty liver, mental attention, motility of the gastrointestinal tract, and exercise performance. 

The data regarding vitamins are more conflicting and controversial, with some evidence suggesting vitamin D might be worth supplementation, and vitamin E was correlated with lower PD risk (Table 9). Interestingly, some types of carotenoids (lutein) were associated with increased risk of PD, but others (β-carotene) were associated with neuroprotection and lower risk of PD. It is worth mentioning that research with vitamins is challenging to obtain useful results for clinical practice due to a significant number of confounding factors.

### 6.4. Smoking

Epidemiological studies have consistently shown that smokers have a lower risk of developing PD, which appears to be dose-dependent, but the reasons for this are not fully understood. Nicotine may stimulate nicotinic acetylcholine receptors in the central nervous system and upregulate brain-derived neurotrophic factors. Noteworthy, while the association between smoking and reduced PD risk is strong, it does not prove causation. Furthermore, the known harmful effects of smoking, including cardiovascular disease, cancer, and respiratory conditions, far outweigh any potential protective effect against PD [191].

Smoking has been inversely associated with Parkinson’s disease (PD) risk, potentially delaying its preclinical and prodromal phases through neuroprotective mechanisms. Nicotine modulates dopaminergic activity, reduces oxidative stress, and suppresses neuroinflammation [192]. Additionally, smoking influences gut microbiota, which may impact α-synuclein aggregation [193]. Understanding these mechanisms could provide insights into neuroprotection strategies.

### 6.5. Sleep and Stress

The association between sleep and the risk of PD is unclear. Patients with high sleep fragmentation rates had more commonly and statistically significant Lewy body pathology (OR 1.40) and substantia nigra neuron loss (OR 1.43) [194]. Another study revealed a statistically significant high risk of developing PD in individuals with poor sleep quality (HR 1.76) and shorter sleep durations (HR 1.72) within six years [195]. In addition, prospective cohort studies and population-based studies suggested that sleep is a cause-reversal effect [196]. Regarding stress, there is one population-based cohort study from Sweden suggesting that high job demand or control may increase the risk of PD [197]. Also, several other studies found that posttraumatic stress disorder was associated with PD risk [198].

### 6.6. Further Work-Up Requirement

There are several neuroimaging and laboratory techniques that can assist in predicting phenoconversion, but outside research population individuals should be evaluated case-by-case. There are significant costs and accessibility to these methods. Some of them require invasive procedures or include risks related to radiation exposure. The results are still imprecise with individual applicability, and uncertain in the results. There is no clear treatment path, or how to interpret the results regarding possible intervention. Moreover, they are associated with a high risk of overdiagnosis with unnecessary treatment, follow-up appointments, and psychological burdens that might influence life choices and relationships.

## 7. Clinical Trials

Prodromal symptoms have been largely studied in different neuropsychological disorders, mainly in Alzheimer’s disease and schizophrenia. The number of studies with prodromal symptoms in synucleinopathies is much lower than other pathologies (Table 10). Moreover, interventions in the prodromal stages of PD have been rarely studied, with most research being observational. A key focus has been on identifying methods to detect patients in the prodromal stage who are at high risk of developing PD. It is noteworthy that the research criteria for prodromal PD were established in 2015, marking an important milestone in efforts to better characterize and intervene during the early stages of the disease. However, further research is needed to explore effective interventions and refine diagnostic approaches for these at-risk individuals.

Clinical trials investigating prodromal synucleinopathies are increasingly focusing on early interventions to delay or prevent the onset of Parkinson’s disease (PD) before motor symptoms manifest. NCT02459886 is examining BIIB054, an anti-alpha-synuclein monoclonal antibody, which targets alpha-synuclein aggregation, a hallmark of PD. Intervening at the prodromal stage could help prevent or reduce alpha-synuclein buildup, potentially altering disease progression. Similarly, trials like NCT04056689 and NCT04760769 are evaluating treatments such as DNL151, which targets the LRRK2 gene associated with PD, and Tavapadon, a dopamine agonist designed to modulate dopaminergic pathways in early PD. These studies are critical in assessing whether early intervention can prevent the onset of motor symptoms and slow disease progression in individuals with early or preclinical signs of synucleinopathies.

In addition to targeting the disease’s biological processes, some trials are focused on the identification of biomarkers to detect PD at its prodromal stage. NCT01141023, the Parkinson’s Progression Markers Initiative (PPMI), is a multi-center cohort study that tracks individuals at risk of developing PD, identifying biomarkers that can signal disease onset before clinical symptoms appear. Similarly, NCT02305147 aims to explore biomarkers that could detect early changes in brain chemistry and imaging, enabling the identification of prodromal PD. The goal of these studies is to improve diagnostic capabilities and identify individuals at risk, facilitating early interventions when therapies are most likely to be effective.

The trials NCT05757206, NCT05934188, NCT03623672, NCT06582121, NCT04724941, NCT06193252, and NCT05611372 represent further efforts to address the early stages of PD and synucleinopathies. These studies focus on evaluating novel drug therapies and biologics that may have the potential to halt or delay disease progression in individuals at risk for PD. These trials aim to explore various therapeutic approaches that target the underlying pathophysiology of synucleinopathies, such as neuroinflammation, synuclein aggregation, and dopaminergic dysfunction, with the ultimate goal of preventing the transition from prodromal stages to full-blown neurodegenerative disease. Together, these studies offer hope for developing treatments that can intervene early and alter the course of Parkinson’s disease before irreversible neuronal damage occurs.

A comprehensive therapeutic approach for synucleinopathies should integrate strategies targeting multiple pathogenic mechanisms, given the complex interplay between α-synuclein aggregation, dopaminergic dysfunction, neuroinflammation, and mitochondrial impairment. While anti-α-synuclein therapies, including monoclonal antibodies and small-molecule inhibitors, aim to reduce misfolded protein accumulation, they may be more effective when combined with neuroprotective agents that enhance mitochondrial function and reduce oxidative stress [199]. Additionally, targeting neuroinflammation through microglial modulation or inflammasome inhibition could mitigate secondary neuronal damage and slow disease progression. Given the differential vulnerability of dopaminergic and non-dopaminergic systems, a multimodal approach incorporating dopamine replacement therapies with interventions preserving noradrenergic and cholinergic function may improve both motor and non-motor symptoms [200]. Future research should explore the synergistic effects of combination therapies to better address the multifactorial nature of synucleinopathies.

## 8. Prediction Algorithms

Prediction algorithms for prodromal synucleinopathies have garnered significant attention due to their potential to identify individuals at high risk before clinical symptoms manifest. These diseases are characterized by the accumulation of alpha-synuclein aggregates, detectable through various biomarkers. Early detection enables timely interventions that may slow disease progression. Traditional diagnostic methods, such as clinical evaluations and imaging, often fail to detect the early stages of synucleinopathies, prompting the exploration of advanced machine learning (ML) models that integrate multiple biomarker types to enhance diagnostic accuracy. These models utilize genetic, neuroimaging, and cerebrospinal fluid (CSF) biomarkers to distinguish prodromal stages of synucleinopathies from healthy controls and other neurodegenerative conditions (Table 11).

Recent studies have demonstrated the efficacy of machine learning models in combining neuroimaging and CSF biomarkers to predict prodromal synucleinopathies. For instance, a study by Koo et al. employed a deep-learning algorithm that utilized both prodromal diagnostic and medication codes to effectively screen for PD [206]. This approach highlights the potential of integrating clinical data with advanced ML techniques to enhance early detection capabilities.

Another significant advancement is the development of stratification tools for disease-modifying trials in prodromal synucleinopathies. A study by Arnaldi et al. investigated whether cost-effective and non-invasive biomarkers could serve as first-line stratification tools [210]. The research emphasizes the importance of identifying reliable biomarkers that can be utilized in clinical settings to predict disease progression and tailor interventions accordingly.

Despite these promising developments, challenges remain in the widespread clinical application of these prediction algorithms. The need for large, diverse datasets that encompass the heterogeneity of synucleinopathies across different stages and subtypes is critical. Additionally, while ML algorithms show great promise, their interpretability remains a significant barrier, as clinicians require clear explanations of model predictions to inform decision-making. Furthermore, external validation across different cohorts and longitudinal studies is essential to assess the generalizability and robustness of these predictive models. As the field evolves, future research must focus on improving model accuracy, reducing the risk of overfitting, and ensuring that these predictive tools can be translated into clinical practice for early detection and personalized treatment of prodromal synucleinopathies.

## 9. Future Studies

More studies should be conducted with different parkinsonian syndromes, especially MSA. The definition of prodromal MSA should be revised and the definition should include parameters for designing the likelihood of developing MSA in the same way as PD. Moreover, it is time for a revision of the criteria for prodromal PD that should include the last ten years of study in PD addressing neuroimaging studies with dopamine transporters and α-synuclein studies. It would be interesting if the Movement Disorder Society could provide online calculators for research in this specific field.

The CSF α-synuclein SAA already showed significant sensitivity and specificity to detect PD, but some individuals still have negative results even with genetic diseases, so further studies evaluating these findings should be performed. These studies should include blood proteome and extracellular vesicle techniques as well as platforms that need to be considered in these cases. Moreover, further progression of new peripheral matrices for biomarkers with reliability and reproducibility is mandatory for the fast development of studies in this field. In this way, other areas that should be investigated are the gut and olfactory mucosa regarding the release of amyloidogenic peptides triggering neurodegeneration.

There have been slowly progressing α-synuclein PET tracer imaging techniques. For the development of the field of prodromal neurodegenerative conditions, this neuroimaging and those also analyzing tauopathies should be developed.

Another field that received attention after the development of Alzheimer’s disease monoclonal therapies is the monoclonal antibodies to sequester α-synuclein in PD. The development of active immunization with AFFITOPE® and the two peptide vaccines (PD01A and PD03A) is ongoing and planned to release results in 2028 (https://affiris.com, 5 February 2025). Regarding passive immunization, Biogen Inc. (Cambridge, MA, USA) showed nonpromising results with cinpanemab [211]. Prasinezumab showed positive results in PASADENA, and ongoing PADOVA results will be released in 2026 (https://www.roche.com, 5 February 2025). AstraZeneca plc (Cambridge, UK) and Takeda Pharmaceutical Company Limited (Tokyo, Japan)developed MEDI-1341 with results planned for the end of 2025 (https://www.astrazenecaclinicaltrials.com, 5 February 2025). AbbVie Inc. (Chicago, IL, USA) withdrew the studies with ABBV-0805 (https://www.bioarctic.com, 5 February 2025). H. Lundbeck A/S (Copenhagen, Denmark) is planning to release the results of LU AF82422 in 2025 (https://www.lundbeck.com, 5 February 2025).

## 10. Conclusions

In sum, the literature about prodromal stages of synucleinopathies is mainly related to PD, with few studies assessing MSA. The development of the MDS criteria for prodromal PD led to significant standardization of the diagnosis and possible evaluation of different therapies in individuals with high risk of PD. However, the literature still lacks therapeutic options for individuals in the prodromal stages of PD, leading to uncertainties and anxiety for these patients. The absence of clear clinical guidelines or effective treatments for this stage of the disease contributes to the distress, as patients face a diagnosis with uncertain implications for their future health. Addressing these gaps through further research and the development of early intervention strategies is crucial to providing better care and support for individuals at risk of PD. Therefore, the assessment of individuals with prodromal stages should still reside in research settings.

α-synuclein SAA can be used as an early diagnostic, but not progression, marker of synuclein disorder. The CSF α-synuclein SAA has different patterns when comparing PD and MSA subjects. Patients with prodromal conditions are more likely to want to know if they can do something to change their disease progression. Among those, the majority of individuals with RBD want to know, and they have high expectations of the healthcare system. Nevertheless, some patients may prefer not to know about their risk of phenoconversion, and in such cases, the physician should respect the patient’s autonomy. It is important for healthcare providers to support patients in making informed decisions based on their preferences, ensuring that discussions around risk and potential outcomes are conducted with sensitivity and consideration for individual wishes.

## Figures and Tables

**Figure 1 clinpract-15-00065-f001:**
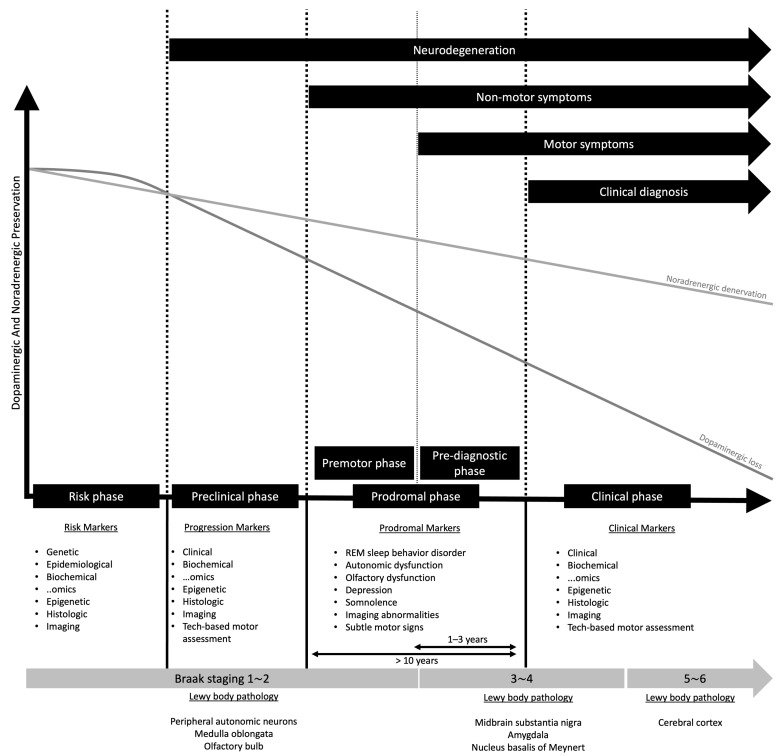
Stages of Parkinson’s disease with biomarkers. Dopaminergic loss and noradrenergic denervation can be observed. Also, the Braak staging and Lewy body pathology can be linked to the stages.

**Figure 2 clinpract-15-00065-f002:**
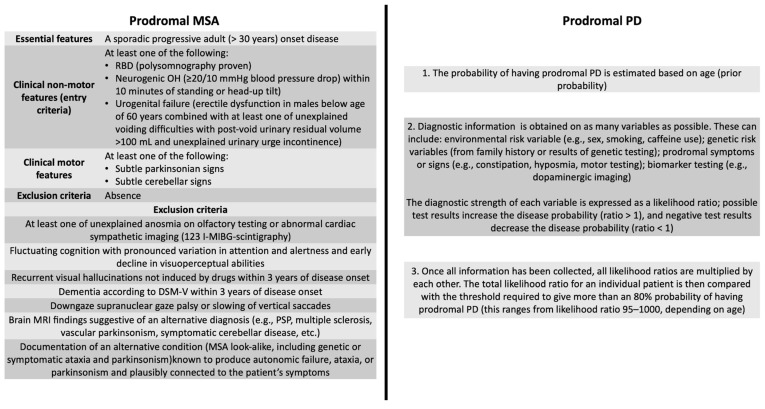
Movement Disorder Society research criteria for prodromal multiple system atrophy (MSA) and Parkinson’s disease (PD). Abbreviations: DMS, Diagnostic and Statistical Manual of Mental; OH, orthostatic hypotension; PSP, Progressive Supranuclear Palsy; RBD, REM sleep behavior disorder.

**Figure 3 clinpract-15-00065-f003:**
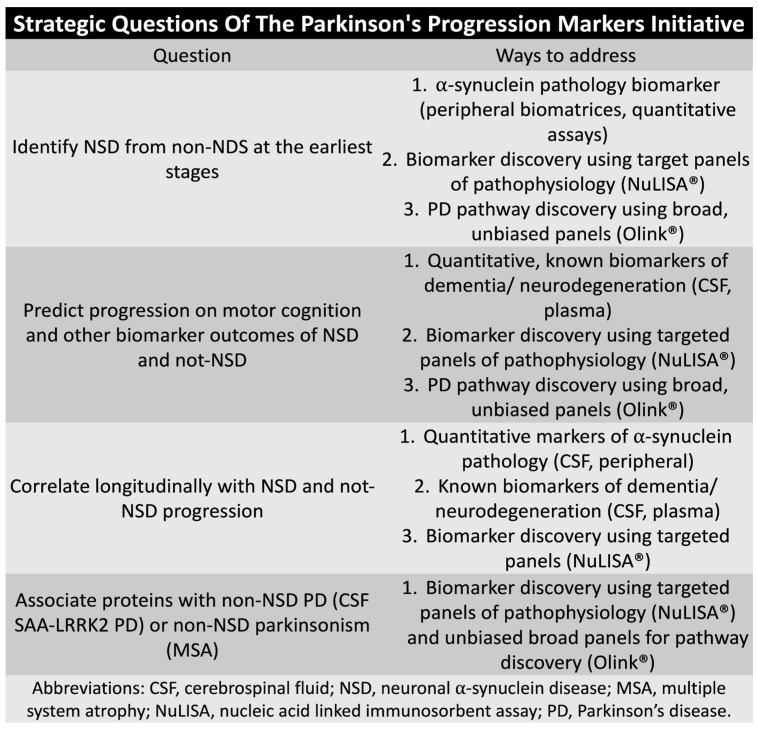
Strategic questions of the Parkinson Progression Marker Initiative (PPMI) study.

**Figure 4 clinpract-15-00065-f004:**
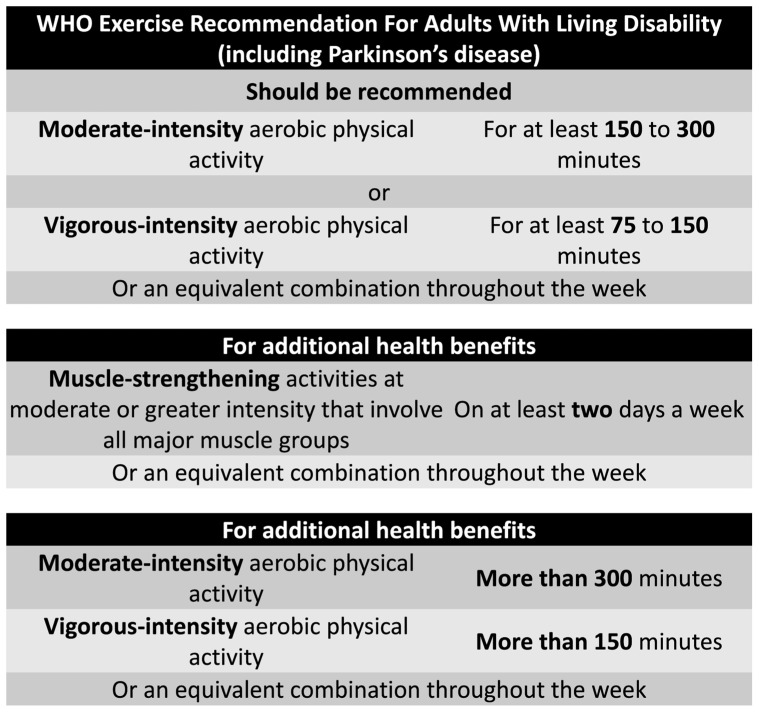
WHO Exercise recommendation for adults with living disabilities (including PD).

**Table 1 clinpract-15-00065-t001:** Studies about prodromal Parkinson’s disease symptoms.

Reference	Year (Start)	Country	Participants	Incident PD	Follow-Up (Years)	Incidence ^a^	Comments
Studies Designed for Investigating Prodromal PD							
Kasten et al. (2013) [7]	2010–continue	Germany (EPIPARK)	715	NA	7	NA	Ongoing study with updates in dataset every two years.
Gaenslen et al. (2014) [8]	2009–2010	Germany (TREND)	698	16	7	3.27	Identified 23 clinical prodromal markers.
Lerche et al. (2014) [9]	NA	Germany and Italy (PRIPS)	1847	21	5	2.27	Patients developing PD after two years have a similar pattern of those with three years.
Jennings et al. (2017) [10]	NA	USA (PARS)	303	26	6	14.30	Hyposmia and abnormal dopamine transporter imaging are predictive of PD conversion.
Hughes et al. (2018) [11]	2012	USA (HFPS/NHS ProPD)	20,726	86	3	1.38	Constipation, RBD, and hyposmia had sensitivity 29% and PPV of 35%.
Mahlknecht et al. (2018) [12]	1990–2005	Italy (Bruneck study)	574	20	10	3.48	MDS prodromal PD has PPV of 78%.
Studies Not Designed for Investigating Prodromal PD							
Ross et al. (2012) [13]	1965	Japan and USA (HAAS)	8000	137	30	0.57	Evaluation of olfactory function, bowel movements, sleep, attention, and executive function are associated with PD.
Hofman et al. (2015) [14]	1990	The Netherlands (Rotterdam study)	14,926	122	31	0.26	NA
Shrestha et al. (2017) [15]	1993	USA (Agricultural health study)	52,394	191	24	0.15	Only in the male sex non-motor symptoms were associated with dose–response to PD.
Healthcare and Claims Databases							
Schrag et al. (2015) [16]	1996	UK (THIN UK)	11 million	8166	10	0.07	Constipation was associated with PD.
Searles Nielsen et al. (2017) [17]	2004	USA (Medicare)	22 million	89,790	5	0.81	Using administrative claims data to predict PD.

Abbreviations: MDS, movement disorder society; NA, not available/not applicable; PD, Parkinson’s disease; PPV, positive predictive value. ^a^ Incidence was calculate with the formula {[(Incident PD/Participants)/Years] × 1000}.

**Table 2 clinpract-15-00065-t002:** Neuroimaging and prodromal parkinsonian syndromes.

Neuroimaging		Descriptions
Dopamine transporter/fluorodeoxyphenylalanine (18F-DOPA)		Striatal dopaminergic denervation
Fluorodeoxyglucose F 18 (18F-FDG)		Disease -related network pattern (PD, LBD), disease-specific metabolism (LBD), and compensatory hypermetabolism (PD)
Technetium Tc 99m hexamethylpropyleneamine oxime single-photon emission computed tomography (99mTc-HMPAO SPECT)		Perfusion changes (prodromal-LBD)
18F-fluoroethoxybenzovesamicol positron emission tomography (18F-FEOBV PET)		Central cholinergic terminal loss (LBD), and possibly cholinergic denervation in the brainstem and pancreas/colon (early LBD study)
Neuromelanin		Loss in the substantia nigra and locus coeruleus
Magnetic resonance imaging	Gray matter	Gray matter atrophy in orbital frontal cortex and amygdaloid body in LBD
	Cortical	Thinning and shape change
	Other	Iron sensitive (T2-weighted), free water (diffusion-weighted), diffusion tensor imaging along the perivascular space
Retinal		Optical coherence tomography and optical coherence tomography angiography
Cardiac (123I-metaiodobenzylguanidine)		Postganglionic sympathetic denervation
Colonic		Transit time and colonic volume

Abbreviations: LBD, Lewy body dementia; PD, Parkinson’s disease.

**Table 3 clinpract-15-00065-t003:** α-Synuclein seeding aggregation assays in prodromal populations.

Reference	Patient Disease (n)	Controls (n)	Matrix	Sensitivity (%)	Specificity (%)	Comment
Fairfoul et al. (2016) [90]	RBD (3)	HC (20)	CSF	100	95	NA
Rossi et al. (2020) [91]	RBD (18)	HC (62)	CSF	100	98	NA
Iranzo et al. (2021) [92]	RBD (52)	HC (40)	CSF	90	90	SAA positivity 10 years before conversion
Stefani et al. (2021) [93]	RBD (63)	HC (40)	OM	44	90	NA
Poggiolini et al. (2022) [94]	RBD (54)	HC (55)	CSF	64	96	NA
Concha-Marambio et al. (2023) [95]	RBD (29)	HC (64)	CSF	93	97	SAA positivity 8.2 years before conversion
Iranzo et al. (2023) [96]	RBD (88)	HC (40)	CSF	75	97	NA
Liguori et al. (2023) [97]	RBD (41)	HC (40)	SB	59	82	NA
Okuzumi et al. (2023) [98]	RBD (9)	HC (128)	Serum	44	91	NA
Siderowf et al. (2023) [99]	RBD (33)	HC (157)	CSF	84	96	NA
	Hyposmia (18)	HC (157)	CSF	88	96	NA
Dam et al. (2024) [100]	RBD (61)	NA	CSF	76	NA	Data from the PPMI, PASADENA, and SPARK studies
	Hyposmia (40)	NA	CSF	72	NA	

Abbreviations: CSF, cerebrospinal fluid; HC, healthy controls; NA, not available/not applicable; RBD, REM sleep behavior disorder; OM, olfactory mucosa; SAA, seed amplification assays; SB, skin biopsy.

**Table 4 clinpract-15-00065-t004:** Potential matrices biomarkers of α-synucleinopathies.

Matrix	Technique	Sensitivity		Specificity	
		Clinical Overt	Prodromal	Clinical Overt	Prodromal
CSF	Oligomeric αSyn	Moderate	Low	Moderate	Moderate
	RT-QuIC αSyn	High	Moderate	High	High
	Total αSyn	Intermediate	Unknown	Low	Unknown
Blood	Oligomeric αSyn	Low	Unknown	Moderate	Unknown
Skin	RT-QuIC αSyn	High	Intermediate	High	High
Olfactory mucosa	RT-QuIC αSyn	Low	Low	High	High
Plasma	RT-QuIC αSyn	High	Intermediate	High	High

Abbreviation: αSyn, α-Synuclein.

**Table 5 clinpract-15-00065-t005:** α-Synuclein in L1CAM-enriched extracellular vesicles (L1EVs) in prodromal populations of patients with REM sleep behavior disorder.

Reference	Sensitivity (%)	Specificity (%)
Niu et al. (2020) [105]	97	54
Jiang et al. (2020) [106]	94	72
Yan et al. (2022) [107]	61	81
Sharafeldin et al. (2023) [108]	69	100
Yan et al. (2024) [104]	86	87

Abbreviations: HC, healthy controls.

**Table 6 clinpract-15-00065-t006:** Lessons learned from Alzheimer’s disease in disclosing amyloid PET results.

Reference	Country	Condition	Participants	Considerations
Lim et al. (2016) [128]	USA	Subjective cognitive decline	11	Disclosure did not affect mood. Those with elevated amyloid were more likely to make lifestyle changes.
Burns et al. (2017) [129]	USA	Cognitive normal	97	No sustained difference in depression and anxiety. Distress predicted by baseline anxiety and depression.
Taswell et al. (2018) [130]	Australia	Mild cognitive impairment	99	No difference in depression and anxiety.
		Alzheimer’s disease	34	
Grill et al. (2020) [131]	USA	Cognitive normal	1705	No difference in depression and anxiety.
Wake et al. (2020) [132]	Japan	Subjective cognitive decline	42	No difference in depression and anxiety.

**Table 7 clinpract-15-00065-t007:** Mediterranean and MIND diets and Parkinson’s disease risk.

Reference	Country	Population	Diet Type	Consideration
Gao et al. (2007) [138]	USA	131,368 HC	MedDiet	RR 0.75
Alcalay et al. (2012) [139]	USA	257 PD, 198 HC	MedDiet	Hight-MedDiet (OR 0.86), and low-MedDiet associated with earlier onset PD
Cassani et al. (2017) [140]	Italy	600 PD, 600 HC	MedDiet and others	No association with PD progression
Mischley et al. (2017) [141]	USA	1053 PD	MedDiet	MedDiet-related foods slow PD progression
Agarwal et al. (2018) [142]	USA	706 HC	MedDiet, MIND, and others	MedDiet HR 0.89 (if adjusted for depression)
Molsberry et al. (2020) [143]	USA	17,400 HC	MedDiet, AHEI, and others	≥3 prodromal features of PD OR 0.82
Paknahad et al. (2020) [144]	Iran	70 PD, 35 HC	MedDiet	MedDiet improves cognitive and motor outcomes
Metcalfe-Roach et al. (2021) [145]	Canada	167 PD, 119 HC	MedDiet, MIND, and others	MedDiet and MIND were associated with later onset of PD
Strikwerda et al. (2021) [146]	Netherlands	9414 HC	MedDiet	HR 0.89
Yin et al. (2021) [147]	Sweden	47,128 HC	MedDiet	HR 0.54 (adjust for age > 65 years-old)
Paknahad et al. (2022) [148]	Iran	70 PD, 34 HC	MedDiet	Motor function improvement
Zhang et al. (2022) [149]	China	71,640 HC	MedDiet	≥2 prodromal features of PD OR 0.74
Lawrie et al. (2023) [150]	UK	162 PD	MIND	No statistically significant effect
Maraki et al. (2023) [151]	Greece	1047 HC	MedDiet	60–70% lower risk for possible/probable prodromal PD
Keramati et al. (2024) [152]	Iran	120 PD and 50 HC	MedDiet	No statistically significant effect

Abbreviations: AHEI, alternative healthy eating index; HC, healthy control; HR, hazard ratio; MedDiet, Mediterranean diet; MIND, Mediterranean-DASH intervention for neurodegenerative delay; OR, odds ratio; PD, Parkinson’s disease; RR, relative risk.

**Table 8 clinpract-15-00065-t008:** Dairy products and Parkinson’s disease risk.

Reference	Country	Population	Consideration
Chen et al. (2002) [156]	USA	135,894 HC	(+) dairy, low-fat milk, cheese; (−) whole milk, yogurt, ice-cream, butter
Park et al. (2005) [157]	Japan	8006 HC	(+) whole and low-fat milk; (−) cheese, ice-cream, butter
Chen et al. (2007) [158]	USA	130,864 HC	(+) dairy, milk, sour cream; (−) cheese, yogurt, ice-cream, butter, cream
Miyake et al. (2010) [159]	Japan	249 PD, 368 HC	(+) None; (−) dairy, milk, cheese, yogurt, ice-cream
Kyrozis et al. (2013) [160]	Greece	26,716 HC	(+) dairy, milk; (−) cheese, yogurt
Sääksjärvi et al. (2013) [161]	Finland	4524 HC	(+) milk, low-fat milk; (−) cheese, yogurt, butter
Jiang et al. (2014) [162]	Meta-analysis		(+) dairy, milk, cheese; (−) yogurt, butter
Hughes et al. (2017) [163]	USA	129,346	(−) low-fat milk
Domenighetti et al. (2022) [164]	European	9823 PD, 368 HC	(+) dairy
Hajji-Louati et al. (2024) [165]	France	71,542 HC	(+) total milk (HR/1-SD 1.09)
Gröninger et al. (2024) [166]	European	183,225 HC	No statistically significant effect

Abbreviations: HC, healthy control; PD, Parkinson’s disease; +, increased PD risk; −, decreased PD risk.

**Table 9 clinpract-15-00065-t009:** Vitamins and Parkinson’s disease risk.

Reference	Country	Population	Vitamin	Consideration
de Rijk et al. (1997) [169]	The Netherland	5342 HC	E	(−): Vit E OR 0.5
Chen et al. (2004) [170]	USA	415 PD	Folate, B6, B12	No statistically significant effect
Etminan et al. (2005) [171]	Meta-analysis	NA	C, E, carotenoids	(−): Vit E RR 0.81 Vit C and carotenoids: no statistically significant effect
de Lau et al. (2006) [172]	The Netherland	5289 HC	Folate, B6, B12	(−): B6 HR 0.69 Folate and B12: no statistically significant effect
Knekt et al. (2010) [173]	Finland	7217 HC	D	(−): Vit D RR 0.33
Miyake et al. (2011) [174]	Japan and UK	249 PD, 368 HC	D	No statistically significant effect
Lv et al. (2014) [175]	Meta-analysis	NA	D	Vit D < 75 nmol/mL (insufficiency): OR 1.5 Vit D < 50 nmol/mL (deficiency): OR 2.2
Takeda et al. (2014) [176]	Meta-analysis	NA	A, carotenoids	No statistically significant effect, except by lutein OR 1.85
Shen et al. (2015) [177]	Meta-analysis	NA	Folate, B6, B12	No statistically significant effect
Hughes et al. (2016) [178]	USA	173,229 HC	β-carotene, C, E	No statistically significant effect
Shrestha et al. (2016) [179]	USA	12,762 HC	D	No statistically significant effect
Luo et al. (2018) [180]	Meta-analysis	NA	D	Vit D 20–30 ng/mL (insufficiency): OR 1.73 Vit D < 20 ng/mL (deficiency): OR 2.08
Wei et al. (2018) [181]	Meta-analysis	NA	E	No statistically significant effect
Ying et al. (2020) [182]	Singapore, China	63,257 HC	A, C, E, carotenoids	No statistically significant effect
Chang et al. (2021) [183]	Meta-analysis	NA	C, E	(−): Vit E OR 0.79 Vit C: no statistically significant effect
Hantikainen et al. (2021) [184]	Sweden	43,865 HC	C, E, β-carotene	(−): Vit C HR 0.68; Vit E HR 0.68 β-carotene: no statistically significant effect
Talebi et al. (2022) [185]	Meta-analysis	NA	C, E, carotenoids	(−): Vit E RR 0.84; Vit C RR 0.94; β-carotene RR 0.94 (+): lutein RR 1.86
Wu et al. (2022) [186]	Meta-analysis	NA	A, β-carotene	(−): β-carotene OR 0.83 Vit A: no statistically significant effect
Flores-Torres et al. (2023) [187]	USA	129,802 HC	Folate, B6, B12	(−): B12 HR 0.80 Folate and B16: no statistically significant effect
Hao et al. (2023) [188]	USA	13,340	E	(−): Vit E OR 0.91
Gröninger et al. (2024) [166]	European	183,225 HC	D	No statistically significant effect
Niu et al. (2024) [189]	Meta-analysis	NA	C, E, β-carotene	(−): Vit E RR 0.87 Vit C and β-carotene: no statistically significant effect
Wang et al. (2024) [190]	European	1.2 million HC	D	No statistically significant effect

Abbreviations: HC, healthy control; HR, hazard ratio; NA, not available/not applicable; OR, odds ratio; PD, Parkinson’s disease; Vit, vitamin; +, increased PD risk; −, decreased PD risk.

**Table 10 clinpract-15-00065-t010:** Clinical trials with prodromal synucleinopathies registered in the ClinicalTrials.gov Database.

Study Start to Completion	Identifier	Condition	Intervention	N Enrolled	Comment
27 September 2018 to 27 April 2020	NCT03671772	RBD	NA	170	Progression of Prodromal Markers of α-synucleinopathy Neurodegeneration in the FDRs of Patients With RBD
June 2010 to 30 June 2020	NCT01141023	PD	DatScan	952	Study to Identify Clinical, Imaging and Biologic Markers of Parkinson Disease Progression (PPMI)
1 September 2021 to 1 September 20212	NCT04266457	RBD, PD, LBD	NA	NA	Establishing Alpha-synuclein RT-QuIC Assay as a Diagnostic Technique in REM Sleep Behaviour Disorder
15 May 2019 to 1 October 2022	NCT04048603	RBD	NA	182	Search for Biomarkers of Neurodegenerative Diseases in Idiopathic REM Sleep Behavior Disorder
1 January 2020 to 1 January 2023	NCT04152655	RBD, PD	Idebenone	180	A Study of Efficacy and Safety of Idebenone vs. Placebo in Prodromal Parkinson Disease (SEASEiPPD)
16 May 2017 to 16 January 2024	NCT05253560	GBA1 Mutation Carriers	NA	600	Prodromal Parkinsonian Features in GBA1 Mutation Carriers
3 January 2022 to 30 June 2024	NCT05353881	RBD	NA	102	Prodromal Markers in Recurrent Dream Enactment Behaviors Without REM Sleep Without Atonia
6 November 2014 to 6 November 2024	NCT02305147	PD	Clinical, biological and imaging follow-up	360	Cohort Study to Identify Predictor Factors of Onset and Progression of Parkinson’s Disease (ICEBERG)
12 August 2022 to 1 May 2025	NCT05826457	LBD, PD, MSA, RBD	NA	500	North American Prodromal Synucleinopathy Consortium Stage 2 (NAPS2)
3 January 2022 to 2 January 2022	NCT05353959	PD	NA	400	Progression Follow up of the First-degree Relatives of Patients with REM Sleep Behavior Disorder
30 April 2021 to March 2025	NCT05677529	PD	NA	8000	Prodromal and Overt Parkinson’s Disease Epidemiological Study in Brazil (PROBE-PD)
10 September 2020 to June 2025	NCT04507139	PD	NA	50	Early Longitudinal Imaging in Parkinson’s Progression Markers Initiative Using [¹⁸F] AV-133 and DaTscan™
4 May 2021 to 1 July 2025	NCT04588285	LBD	Ambroxol	180	Ambroxol in New and Early DLB, A Phase IIa Multicentre Randomized Controlled Double Blind Clinical Trial (ANeED)
15 September 2022 to 31 August 2025	NCT05757206	RBD	Syn-One Test	80	The Syn-Sleep Study
1 May 2023 to 30 April 2026	NCT05934188	PD	NA	200	Exploring the Gut–Brain Axis in Ageing and Neurodegeneration (GutBrain)
29 August 2018 to 31 July 2026	NCT03623672	LBD, PD, MSA, RBD	NA	500	North American Prodromal Synucleinopathy (NAPS) Consortium
October 2024 to November 2026	NCT06582121	RBD, PD	Polysomnography	457	Study of Sleep Disorders in Prodromal and Definite Parkinsons Disease (SOMPARK)
1 July 2021 to December 2026	NCT04724941	PD	NA	2000	Prodromal Alpha-Synuclein Screening in Parkinson’s Disease Study (PASS-PD)
15 January 2024 to 1 December 2026	NCT06193252	PD, RBD	Physical activity	110	Slowing Parkinson’s Early Through Exercise Dosage-Netherlands (Slow-SPEED-NL)
1 January 2023 to 31 December 2026	NCT05611372	RBD, PD	Rasagiline	732	Efficacy and Safety of Rasagiline in Prodromal Parkinson’s Disease
12 April 2024 to 31 December 2026	NCT06456684	PD	Fluoro [18F]promethazine	76	AV133 Longitudinal Imaging Study in Patients With Early and Prdromal Parkinson’s Disease
8 February 2024 to 30 December 2028	NCT06467461	LBD, PD, RBD	Skin biopsy, speech testing, ultra-high field 7T MRI	60	Identification of Prodromal Neurodegeneration in Serotonergic-Induced REM Sleep Behavior Disorder
1 April 2024 to 30 December 2028	NCT06420310	PD	Esposure to pesticide	260	Pesticides and Parkinson’s Disease (Pest-PD)
1 February 2023 to December 2032	NCT05740683	Anosmia, hyposmia, olfactory dysfunction	RT-QuiC	100	Alpha-synuclein Rt-quic and Neurologic Symptoms in Persons With idiOpathic anosMiA (AROMA)
1 July 2020 to December 2033	NCT04477785	PD	NA	4500	PPMI Clinical—Establishing a Deeply Phenotyped PD Cohort
28 July 2021 to December 2041	NCT05065060	PD	NA	500,000	Parkinson Progression Marker Initiative Online (PPMI Online)

Abbreviations: LBD, Lewy body disease; PD, Parkinson’s disease; RBD, REM sleep behavior disorder.

**Table 11 clinpract-15-00065-t011:** Machine learning models for predicting prodromal Parkinson’s disease.

Study	N	Type of Data	Machine Learning Models	Technique	Key Findings	Comparison with Other Models	AUC, SN, SP
Karabayir et al. (2023) [201]	1189	ECG data	Deep Learning Model	Convolutional Neural Networks	Developed a deep learning model to identify prodromal PD with high accuracy from ECG data.	Compared with logistic regression (ML), outperforming it.	AUC 0.74
Vaish et al. (2024) [202]	NA	Clinical Data	Machine Learning Approach	NA	Developed an ML prediction model to improve risk prediction for PD, enabling early intervention and resource prioritization.	NA	NA
Warden et al. (2021) [203]	88,265	Administrative claims data	Various Prediction Approaches	Logistic Regression, Random Forest	Compared different ML-based prediction approaches for identifying prodromal PD using claims data.	Compared multiple ML models (logistic regression, decision trees, random forest).	Combined approach was the best model with AUC 0.83; SN 0.76; SP 0.76
Tabashum et al. (2024) [204]	NA	Various data sources	Systematic Review of ML Models	Multiple Techniques	Systematic review highlighting the effectiveness of ML in predicting PD, but with variation in reported metrics.	Compared multiple ML models (overview study).	NA
Makarious et al. (2022) [205]	PPMI study	Multimodal data (genetic, clinical)	Automated ML Framework (GenoML)	Ensemble Learning	Multimodal model combining genetic and clinical data to predict PD risk systematically.	Compared with single-modality models, demonstrating superior accuracy.	AUC 0.85; SN 0.93; SP 0.43
Koo et al. (2025) [206]	9020	Diagnostic and medication codes	Deep Learning Algorithm	Recurrent Neural Networks	Developed a deep learning model using diagnostic and medication data to screen for prodromal PD.	Compared with traditional statistical models, outperforming them.	AUC 0.92; SN 0.81; SP 0.94
Prashant et al. (2018) [207]	PPMI study	Patient Questionnaire Data	Logistic Regression, Random Forests, Boosted Trees, SVM	Supervised Learning	Developed models to classify early PD from healthy controls using patient questionnaire data.	Compared multiple ML models (SVM, boosted trees, logistic regression).	SVM was the best model with AUC 0.96–0.98; SN 0.95–0.97; SP 0.82–0.94. But all the models had AUC from 0.96 to 0.98
Dehsarvi et al. (2019) [208]	128	Resting-State fMRI Data	Evolutionary Algorithms	Cartesian Genetic Programming	Developed automatic methods for detecting brain imaging preclinical biomarkers for PD, achieving high classification accuracies.	Compared with Artificial Neural Networks (ANN) and Support Vector Machines (SVM); CGP provided comparable performance.	SN was 0.75 for differentiating prodromal PD from healthy controls
Tran et al. (2023) [209]	296	Retinal Fundus Imaging	Deep Learning Models	Transfer Learning	Predicted prevalent and incident PD from fundus imaging using deep learning.	Compared with conventional feature extraction models, showing improvement.	AlexNet was the best model with AUC 0.77; SN 0.76; SP 0.60

Abbreviations: AUC, area under the curve; ECG, electrocardiogram; ML, machine learning; N, Number of participants in the study; NA, not applicable/not available; PD, Parkinson’s disease; SN, sensitivity; SP, specificity; SVM, support vector machines.

## Data Availability

No new data created.

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
