# Peer review of "A Narrative Review on Biochemical Markers and Emerging Treatments in Prodromal Synucleinopathies"

_clinpract, 2025, doi:10.3390/clinpract15030065_

Round 1
Reviewer 1 Report
Comments and Suggestions for Authors
This manuscript “A Narrative Review on Biochemical Markers and Emerging Treatments in Prodromal Synucleinopathies” is well-structured and comprehensive, with a clear focus on the prodromal stage of synucleinopathies. The authors effectively provide a broad yet detailed overview of clinical, neuroimaging, and biochemical markers relevant to early Parkinson’s disease (PD) and other disorders like multiple system atrophy (MSA), as well as practical guidance for patients and caregivers. The manuscript is divided into clearly defined sections, each addressing a key aspect of prodromal synucleinopathies. This well-designed organization helps guide the reader through complex information. The narrative is clear and engaging, making it accessible and informative to a wide audience of clinicians, researchers, and healthcare professionals.
While the overall construction and narrative of the paper is in very good shape, there are some minor suggestions for improvement:
- The figures in the paper can be in higher resolution. especially for Figure 4, The link under the box is vague and unable to read.
- In Table 10. The title contains a typo, “Synucleionopathies” should be “Synucleinopathies”.
Author Response
This manuscript “A Narrative Review on Biochemical Markers and Emerging Treatments in Prodromal Synucleinopathies” is well-structured and comprehensive, with a clear focus on the prodromal stage of synucleinopathies. The authors effectively provide a broad yet detailed overview of clinical, neuroimaging, and biochemical markers relevant to early Parkinson’s disease (PD) and other disorders like multiple system atrophy (MSA), as well as practical guidance for patients and caregivers. The manuscript is divided into clearly defined sections, each addressing a key aspect of prodromal synucleinopathies. This well-designed organization helps guide the reader through complex information. The narrative is clear and engaging, making it accessible and informative to a wide audience of clinicians, researchers, and healthcare professionals. While the overall construction and narrative of the paper is in very good shape, there are some minor suggestions for improvement:
The figures in the paper can be in higher resolution. especially for Figure 4, The link under the box is vague and unable to read.
Authors: Corrected, and images size and quality increased.
In Table 10. The title contains a typo, “Synucleionopathies” should be “Synucleinopathies”.
Authors: corrected
Reviewer 2 Report
Comments and Suggestions for Authors
The manuscript covers key aspects of prodromal synucleinopathies, including biomarkers, neuroimaging markers, and risk factors. In terms of English language, the text is fairly clear, and I didn’t find major errors; however, I think that in some parts, the sentence structure could be modified, as some sentences are somewhat rigid and complex. These small adjustments will certainly improve the manuscript’s readability.
I believe the introduction could be improved. I would ask the authors to expand on it, at least providing the basics of PD, a definition, and everything that would certainly be useful for the readers to understand the disease. This is because the review may be read by students or a less expert audience.
The following chapter (2) is well-structured, and it explains what prodromal synucleinopathy in Parkinson’s disease is. This is important for a less experienced audience. In this section of the review, I would also recommend increasing the font size in figure 1, as it may appear small.
Also in this chapter, I think the authors could go deeper into the discussion of the pathophysiology of RBD and PD. Adding a brief discussion on the molecular and cellular biology behind these changes could make the text more complete.
Chapter 3 mentions that dopamine loss is an important indicator of disease progression, but it would be interesting to delve deeper into the type of dopamine that is lost and how these losses are associated with specific motor and non-motor symptoms.
I think that in the central chapters of the manuscript, although very well-developed and treated, the molecular biological phenomena underlying the disease should be addressed in greater detail. I understand that this is a clinical review, but I believe it could be helpful.
In the final part of the text, particularly in chapter 7, many of the studies mentioned focus on treatments targeting specific pathogenic mechanisms, such as alpha-synuclein aggregation or dopaminergic dysfunction. It would be interesting to discuss how these approaches could be combined to address the complexity of synucleinopathies, which involve different neurodegenerative pathways.
In terms of English language, the text is fairly clear, and I didn’t find major errors; however, I think that in some parts, the sentence structure could be modified, as some sentences are somewhat rigid and complex. These small adjustments will certainly improve the manuscript’s readability.
Author Response
The manuscript covers key aspects of prodromal synucleinopathies, including biomarkers, neuroimaging markers, and risk factors. In terms of English language, the text is fairly clear, and I didn’t find major errors; however, I think that in some parts, the sentence structure could be modified, as some sentences are somewhat rigid and complex. These small adjustments will certainly improve the manuscript’s readability.
I believe the introduction could be improved. I would ask the authors to expand on it, at least providing the basics of PD, a definition, and everything that would certainly be useful for the readers to understand the disease. This is because the review may be read by students or a less expert audience.
Parkinson's disease (PD) is a progressive neurodegenerative disorder primarily characterized by motor symptoms such as bradykinesia, resting tremor, rigidity, and postural instability. These clinical manifestations are attributed to the degeneration of dopaminergic neurons in the substantia nigra pars compacta, leading to striatal dopamine deficiency. Beyond motor symptoms, PD encompasses a range of non-motor fea-tures—including hyposmia, constipation, rapid eye movement (REM) sleep behavior disorder (RBD), depression, and cognitive impairment—that often precede motor signs by years, indicating a prodromal phase of the disease.
PD is classified within the spectrum of synucleinopathies, a group of neurodegen-erative disorders marked by the pathological accumulation of misfolded alpha-synuclein protein in neurons, glial cells, or nerve fibers. This group also includes dementia with Lewy body dementia (LBD) and multiple system atrophy (MSA), each presenting distinct clinical profiles yet sharing overlapping pathological features. In PD and LBD, al-pha-synuclein aggregates predominantly in neurons, forming Lewy bodies and Lewy neurites, whereas in MSA, the protein accumulates mainly in oligodendrocytes as glial cytoplasmic inclusions [1].
Historically, research has predominantly focused on the symptomatic treatment of PD during its clinical phase. However, over the past decade, there has been a paradigm shift towards understanding the prodromal stages of PD and other synucleinopathies. This shift aims to identify early biomarkers and therapeutic targets to facilitate dis-ease-modifying interventions before the onset of overt clinical symptoms. Notably, markers such as olfactory loss, RBD, and constipation have been identified as significant predictors of prodromal PD [2].
In this review, we critically examine the current landscape of clinical, biochemical, and neuroimaging markers pertinent to the prodromal phases of synucleinopathies. We also discuss strategies for counseling and managing individuals identified as being in these early stages, with the objective of providing a comprehensive resource for movement disorder specialists and researchers dedicated to advancing the early diagnosis and intervention in synucleinopathies.
The following chapter (2) is well-structured, and it explains what prodromal synucleinopathy in Parkinson’s disease is. This is important for a less experienced audience. In this section of the review, I would also recommend increasing the font size in figure 1, as it may appear small.
Author: The authors would like to request to maintain the current figure letter size to ensure readability, as the figure contains a substantial amount of information.
Also in this chapter, I think the authors could go deeper into the discussion of the pathophysiology of RBD and PD. Adding a brief discussion on the molecular and cellular biology behind these changes could make the text more complete.
Authors: RBD and PD share a common molecular pathology centered on α-synuclein misfolding and aggregation. In RBD, early deposition of phosphorylated α-synuclein occurs in brainstem nuclei, particularly the sublaterodorsal nucleus and magnocellular reticular formation, leading to dysfunction in REM sleep atonia pathways. This pathology spreads via prion-like mechanisms, progressively affecting dopaminergic neurons in the sub-station nigra pars compacta [19]. At the cellular level, α-synuclein aggregates impair mitochondrial function, disrupt endoplasmic reticulum (ER) homeostasis, and activate neuroinflammatory pathways through microglial and astrocytic activation. Dysregulated protein degradation via the ubiquitin-proteasome system and autophagy-lysosomal pathways exacerbates neuronal stress, contributing to progressive neurodegeneration [20]. Additionally, tau co-pathology and lysosomal dysfunction further accelerate disease progression. Identifying molecular signatures of α-synuclein pathology in RBD may facilitate early diagnosis and neuroprotective interventions [21].
Chapter 3 mentions that dopamine loss is an important indicator of disease progression, but it would be interesting to delve deeper into the type of dopamine that is lost and how these losses are associated with specific motor and non-motor symptoms.
Authors: Dopamine loss in PD is not uniform; it involves distinct nigrostriatal, mesocortico-limbic, and tuberoinfundibular pathways, each contributing to specific motor and non-motor symptoms. In the early stages, PD primarily affects dorsal striatal dopamin-ergic terminals, particularly the putamen, leading to bradykinesia and rigidity due to impaired D1/D2 receptor-mediated modulation of the direct and indirect basal ganglia pathways [41]. As the disease progresses, ventral striatal dopamine loss disrupts reward processing, contributing to apathy and depression. Degeneration of mesocortical dopamine neurons projecting to the prefrontal cortex impairs executive function and working memory, while alterations in the tuberoinfundibular pathway contribute to hyperprolactinemia and autonomic dysfunction [42]. Additionally, differential loss of tonic versus phasic dopamine release exacerbates motor fluctuations and cognitive deficits [43].
I think that in the central chapters of the manuscript, although very well-developed and treated, the molecular biological phenomena underlying the disease should be addressed in greater detail. I understand that this is a clinical review, but I believe it could be helpful.
In the final part of the text, particularly in chapter 7, many of the studies mentioned focus on treatments targeting specific pathogenic mechanisms, such as alpha-synuclein aggregation or dopaminergic dysfunction. It would be interesting to discuss how these approaches could be combined to address the complexity of synucleinopathies, which involve different neurodegenerative pathways.
Authors: A comprehensive therapeutic approach for synucleinopathies should integrate strategies targeting multiple pathogenic mechanisms, given the complex interplay between α-synuclein aggregation, dopaminergic dysfunction, neuroinflammation, and mitochondrial impairment. While anti-α-synuclein therapies, including monoclonal antibodies and small-molecule inhibitors, aim to reduce misfolded protein accumulation, they may be more effective when combined with neuroprotective agents that enhance mitochondrial function and reduce oxidative stress [199]. Additionally, targeting neuroinflammation through microglial modulation or inflammasome inhibition could mitigate secondary neuronal damage and slow disease progression. Given the differential vulnerability of dopaminergic and non-dopaminergic systems, a multimodal approach incorporating dopamine replacement therapies with interventions preserving noradrenergic and cholinergic function may improve both motor and non-motor symptoms [200]. Future research should explore the synergistic effects of combination therapies to better address the multifactorial nature of synucleinopathies.
Reviewer 3 Report
Comments and Suggestions for Authors
The introduction has to clearly state the primary research questions and objectives. The present version lacks a strong thesis statement and fails to adequately define the scope of the review.
Certain arguments lack appropriate citations, whereas others depend excessively on a restricted number of studies in table.
The evaluation is primarily descriptive. A comprehensive evaluation of the advantages and drawbacks of various biomarkers, neuroimaging methodologies, and treatment approaches is necessary.
The publication must incorporate a section addressing the limitations and obstacles associated with the implementation of these biomarkers in clinical practice, including factors such as cost, availability, and standardization.
The conclusion fails to sufficiently encapsulate essential findings or provide avenues for future investigation. A greater emphasis on the potential of these biomarkers to facilitate earlier intervention measures is required.
Author Response
Reviewer 3
The introduction has to clearly state the primary research questions and objectives. The present version lacks a strong thesis statement and fails to adequately define the scope of the review.
Authors: It is a narrative review, as the title mentions. Also, we modify the aim to improve clarity. “In this review, we critically examine the current landscape of clinical, biochemical, and neuroimaging markers pertinent to the prodromal phases of synucleinopathies. We also discuss strategies for counseling and managing individuals identified as being in these early stages, with the objective of providing a comprehensive resource for movement disorder specialists and researchers dedicated to advancing the early diagnosis and intervention in synucleinopathies.”
Certain arguments lack appropriate citations, whereas others depend excessively on a restricted number of studies in table.
Authors: please reviewer mark these errors, and we will likely improve the quality of the manuscript. Also, consider including citations according to the references.
The evaluation is primarily descriptive. A comprehensive evaluation of the advantages and drawbacks of various biomarkers, neuroimaging methodologies, and treatment approaches is necessary.
Authors: There is no current data to support any hypothesis; the only way would be to do a systematic review with a meta-analysis of all the studies mentioned, which unfortunately goes beyond the aim of the current manuscript but is something that should be done in the future. Also, based on this assumption, we include the chapter about counseling. There is no current guideline about prodromal phases.
The publication must incorporate a section addressing the limitations and obstacles associated with the implementation of these biomarkers in clinical practice, including factors such as cost, availability, and standardization.
Authors: “Chapter 6.6. Further work-up requirement” has information about these limitations.
The conclusion fails to sufficiently encapsulate essential findings or provide avenues for future investigation. A greater emphasis on the potential of these biomarkers to facilitate earlier intervention measures is required.
Authors: the information regarding future studies were included in the chapter before the conclusion “9. Future Studies.”
We appreciate the Reviewer’s comments and value their feedback. However, we kindly request a revised version of the review with more specific references to lines or chapters, along with concrete suggestions to enhance the manuscript’s quality. Additionally, if the Reviewer is an expert in the field, we would greatly appreciate relevant citations that could further strengthen our discussion. Given that the current manuscript includes over 200 references, we believe we have comprehensively covered the literature on the prodromal phase. Nevertheless, we are open to considering any key studies that may have been overlooked.
Round 2
Reviewer 2 Report
Comments and Suggestions for Authors
The authors have improved their article, and it has a good level of interest and significance, making it appropriate for this journal. I believe it can be considered by the Editors of this journal.
Reviewer 3 Report
Comments and Suggestions for Authors
The author changed nicely.